# Trial of intraoperative cell salvage versus transfusion in ovarian cancer (TIC TOC): protocol for a randomised controlled feasibility study

Khadra Galaal,[1] Alberto Lopes,[2] Colin Pritchard,[3] Andrew Barton,[4] Jennifer Wingham,[5] Elsa M R Marques,[6] John Faulds,[7] Joanne Palmer,[3] Patricia Jane Vickery,[8] Catherine Ralph,[9] Nicole Ferreira,[7] Paul Ewings[4]

For numbered affiliations see end of article.

**Correspondence to**
Miss Khadra Galaal;
k.galaal@nhs.net

## ABSTRACT

**Introduction** Ovarian cancer is the leading cause of death from gynaecological cancer, with more than 7000 new cases registered in the UK in 2014. In patients suitable for surgery, the National Institute of Health and Care Excellence guidance for treatment recommends surgical resection of all macroscopic tumour, followed by chemotherapy. The surgical procedure can be extensive and associated with substantial blood loss which is conventionally replaced with a donor blood transfusion. While often necessary and lifesaving, the use of donor blood is associated with increased risks of complications and adverse surgical outcomes. Intraoperative cell salvage (ICS) is a blood conservation strategy in which red cells collected from blood lost during surgery are returned to the patient thus minimising the use of donor blood. This is the protocol for a feasibility randomised controlled trial with an embedded qualitative study and feasibility economic evaluation. If feasible, a later definitive trial will test the effectiveness and cost-effectiveness of ICS reinfusion versus donor blood transfusion in ovarian cancer surgery.

**Methods and analysis** Sixty adult women scheduled for primary or interval ovarian cancer surgery at participating UK National Health Service Trusts will be recruited and individually randomised in a 1:1 ratio to receive ICS reinfusion or donor blood (as required) during surgery. Participants will be followed up by telephone at 30 days postoperatively for adverse events monitoring and by postal questionnaire at 6 weeks and 3 monthly thereafter, to capture quality of life and resource use data. Qualitative interviews will capture participants' and clinicians' experiences of the study.

**Ethics and dissemination** This study has been granted ethical approval by the South West–Exeter Research Ethics Committee (ref: 16/SW/0256). Results will be disseminated via peer-reviewed publications and will inform the design of a larger trial.

**Trial registration number** ISRCTN19517317.

## Strengths and limitations of this study

► This is the first study to use intraoperative cell salvage in cytoreductive surgery for ovarian cancer.
► The study explores the feasibility and informs the design of a larger randomised controlled trial. Quantitative, qualitative and feasibility economic components are included.
► The effect of transfusion and cell salvage on immune response to surgery is not assessed.
► This feasibility study will not provide information on the long-term outcomes of using either cell salvage or transfusion.

## INTRODUCTION
### Background

Ovarian cancer is the leading cause of death from gynaecological cancer in the UK (age-standardised mortality rate 9.1 per 100 000, 2008–2010).[1] Although survival rates have improved in recent decades, there are still more deaths from ovarian cancer than all other gynaecological cancers combined.[2] The mainstays of treatment for advanced ovarian cancer are surgical cytoreduction and platinum-based chemotherapy. As operative success and survival is largely determined by residual disease.[3] Surgery is often extensive with substantial intraoperative blood loss, about 53% of patients lose more than 1.5 L during their first surgery.[4] Blood lost during surgery is conventionally replaced using donor blood transfusion with the incidence of transfusion ranging from 35% to 77%.[5 6] Perioperative donor blood transfusion is associated with increased risks of complications and adverse surgical outcomes including mortality, wound infection, pulmonary and renal complications, systemic sepsis and prolonged hospital stay.[7] In 2012, there were 12.3 serious adverse incidents per 10 000 transfused components reported by the Serious Hazards of Transfusion (SHOT) group.[8] SHOT is an independent, professionally led scheme, involved in collecting and analysing anonymised information on adverse events and reactions in

blood transfusion from all healthcare organisations in the UK. Where risks and problems are identified, they produce recommendations to improve patient safety. One suggested explanation for adverse reactions is a general transient depression of the immune system following transfusion with blood products, transfusion-induced immunomodulation (TRIM).[9 10]

Intraoperative cell salvage (ICS) or autologous blood transfusion is the practice of recovering red cells from blood lost in the operative field and returning them to the patient.[11] This process involves the separation, centrifugation, washing and filtration of heparinised red blood cells, before reinfusion into the patient. ICS eliminates or reduces the need for donor blood transfusion and its associated risks, making it an alternative where major blood loss is anticipated.[12] ICS can be available in theatre at modest expense and reduces dependence on the limited pool of banked blood. Studies comparing cell salvage with allogeneic blood transfusion (ABT) have demonstrated increased mean erythrocyte (red blood cells) viability as high as 88% with cell salvage.[13–15] ICS has been used successfully in surgical specialties[16] including cardiothoracic, vascular, orthopaedic and hepatobiliary.[17–20] In addition, ICS is associated with low rate of patient-related adverse events.[21] ICS was initially contraindicated in cancer because of the theoretical risk of reintroducing malignant tumour cells into patients' bloodstreams.[22 23] However, such concerns appear to be unfounded.[24] The in vitro leucocyte depletion filters are highly efficient at removing malignant cells with removal rates of between 80% and 100%.[25 26] In patients undergoing surgery for gynaecological malignancy, leucocyte depletion filters effectively eliminate viable nucleated malignant cells from the returned blood.[27 28] Far from compromising outcomes, ICS is associated with improved outcomes in cervical[29 30] and oesophageal cancers.[24]

Interestingly, patients with primary metastatic cancer are known to have circulating tumour cells (CTC) in the blood.[31] Furthermore, operative manipulation of tumours during surgery leads to peripheral blood concentrations of malignant cells many times higher than could be attained with cell salvage.[32] The presence of CTC is prevalent in patients with cancer with approximately one CTC per 105 to 107 mononuclear cells found in the peripheral blood of patients with metastatic cancer.[33]

## Rationale

There is a paucity of studies in ICS, making it difficult for patients, clinicians and National Health Service (NHS) managers to make decisions about this technology.[34] ICS has been used in patients with ovarian cancer in one of the participating sites with encouraging results, but a randomised controlled trial (RCT) is required for robust determination of effectiveness. The aim of a definitive trial would be to assess the clinical and cost-effectiveness of ICS for women undergoing cytoreductive surgery for ovarian cancer, compared with usual practice of transfusing only allogeneic blood as required.

## Aim and objectives

The aim of the study is to determine whether a definitive RCT is feasible and, if so, how best to deliver it. The objectives of the study are to:

► Estimate the likely recruitment rate for the larger trial
► Estimate the likely completeness of resource use and outcome data
► Explore the practical logistics of undertaking randomisation in theatres
► Assess success of blinding of allocation for participants and outcome assessors
► Design data collection tools to collect resource use data from participants, hospital medical records and hospital staff
► Inform the trial design and confirm the resources required to run a larger definitive trial
► Explore the barriers and facilitators for women when deciding whether or not to participate
► Explore women's perceptions of:
  – The intervention, the information given and advantages/disadvantages of participation so that information can be optimised for the larger trial
  – Other trial aspects, for example, regarding collection of outcome measures and completing resource use questionnaires.
► Identify factors influencing surgeons' decisions about whether or not to participate in the study.

## METHODS
### Trial design

This is a protocol for a randomised, controlled, multicentre feasibility study in women undergoing cytoreductive surgery for ovarian cancer. Sixty participants will be individually randomised in a 1:1 ratio to ICS (reinfusion of their own blood) or donor blood transfusion during surgery. Participants and outcome assessors will be blinded to the intervention. All participants will be followed up by telephone for adverse events reporting at 30 days postoperatively, by post 6 weeks postoperatively and 3 monthly thereafter as time allows. A schematic diagram of the trial is given in figure 1. The feasibility study includes an embedded qualitative component to assess participants' (patients and clinicians) perceptions of their experience in preparation for the later trial. It will also involve an assessment of the feasibility of collecting resource use and other economic data for a future economic evaluation.

### Study setting

The study will take place at the Royal Cornwall Hospitals NHS Trust, Plymouth Hospitals NHS Trust, Gateshead Health NHS Foundation Trust and University Hospitals of Leicester NHS Trust. All sites have existing personnel experienced in the management of ICS and reinfusion.

### Participants and recruitment

Participants will be recruited from patients scheduled to undergo surgery for ovarian cancer at the participating hospitals. Potential participants will usually be

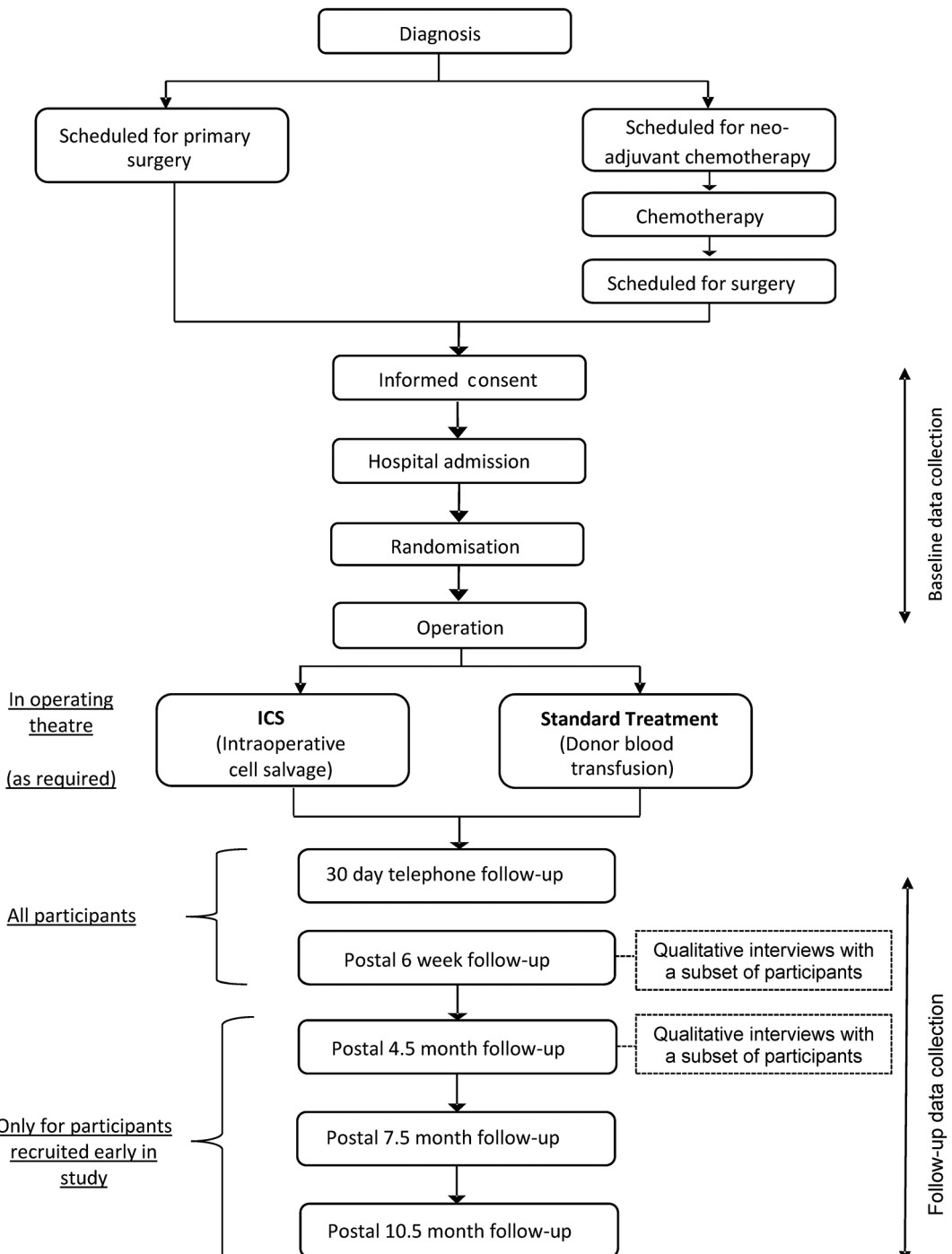

**Figure 1** Summary of trial design. ICS, intraoperative cell salvage .

identified from those patients attending the gynaecological oncology outpatient clinic having been referred by their general practitioner under the 2-week wait cancer pathway. Some patients will be scheduled for primary surgery and are suitable for immediate recruitment to the study. Others will undergo neoadjuvant chemotherapy prior to interval debulking surgery and may be recruited to the study at a later date, following chemotherapy. Written informed consent (see online supplementary appendices 1 and 2) will be obtained by an appropriately trained member of the research team in line with

INTERNATIONAL CONFERENCE ON HARMONISATION (ICH) Good Clinical Practice guidelines. As part of the consent process, patients will be reminded that they are free to withdraw from the study at any time without giving a reason and without affecting their future treatment.

### Inclusion criteria
Potential participants must satisfy the following criteria to be enrolled in the study:
▶ 18 years old or over

- ► Suspected or confirmed ovarian cancer (newly diagnosed) requiring cytoreductive surgery, whether primary or interval (following chemotherapy)
- ► CT scan evidence (with or without clinical evidence) compatible with FIGO stage III/IV ovarian cancer/primary peritoneal cancer at presentation[35] (see online supplementary appendix 3)
- ► Eastern Cooperative Oncology Group performance status 0–1[36]
- ► Willing to participate and able to give written informed consent

### Exclusion criteria

Potential participants meeting any of the following criteria will be excluded from study participation:
- ► Diagnosis of concurrent malignancy
- ► Pregnant
- ► Haemoglobinopathies (eg, sickle cell, thalassaemia)
- ► Unwilling to accept donor blood (eg, on religious grounds)

### Randomisation

Randomisation will be undertaken after written consent has been obtained, but as close to the start of surgery as possible; usually, this will be on the morning of the operation day but if this is not possible for practical reasons, it may be performed earlier. Randomisation will be achieved by means of a web-based system created by the UK Clinical Research Collaboration-registered Peninsula Clinical Trials Unit (CTU) in conjunction with the trial statistician, using random permuted blocks of varying size. Participants will be allocated to receive ICS reinfusion or donor blood transfusion in a 1:1 ratio, stratified by study site. To prevent any unnecessary delays in the operating theatre, cell salvage equipment will be set up in advance for all study participants, before confirmation of treatment allocation.

### Trial interventions

Participants will be allocated to receive either donor blood transfusion or ICS reinfusion intraoperatively, in accordance with specified transfusion protocols. Donor blood will only be given (in standard volumes) when deemed necessary (eg, after substantial blood loss and/or drop in haemoglobin) whereas ICS blood will be returned even if only small quantities are lost. Some participants may not require any intraoperative transfusion and some (in either arm of the trial) may require donor blood transfusion postoperatively.

### Intraoperative cell salvage

All sites will follow a common ICS protocol and relevant site staff will undergo study-specific training prior to the study start. Collected blood will be processed via the ICS machine before being reinfused via a leucodepletion filter. The make and model of ICS machine and leucodepletion filter used in clinical practice varies across NHS Trusts and will not be standardised for this feasibility study. Relevant data from a local intraoperative cell

salvage audit form will be transcribed into the study-specific case report form (CRF), including the amounts of salvaged blood processed and reinfused.

### Donor transfusion

Participants allocated to donor transfusion will be considered for transfusion during surgery in accordance with clinical judgement, guided by local hospital policy. The factors triggering transfusion (eg, excessive blood loss, hypotension, reduced Hb) will be documented in the CRF along with the amount and type of blood and blood products transfused.

### Donor transfusion in ICS arm

Participants allocated to the ICS arm who need donor transfusion can be given donor blood at any time, during or after surgery, for the duration of their hospital stay. The factors triggering intraoperative donor transfusion in the ICS group will be documented in the CRF as well as the amount and type of any blood and blood products transfused.

### Blinding

Surgeons, other theatre staff and the person recording details of intraoperative blood transfusion or reinfusion cannot be blinded in this study. The research nurse responsible for recording postoperative outcomes will aim to remain blinded to treatment allocation. Participants in either arm of the study may have some form of blood replacement in progress immediately postsurgery; it is unlikely that participants will be able to distinguish between the two types and either group may require donor blood for clinical reasons.

### Feasibility outcomes

The outcomes for this study are the feasibility and acceptability of the study and study procedures in relation to recruitment, randomisation, intervention, blinding, participant retention and data completion. Both quantitative and qualitative methods will be used. Recruitment rate will be measured as the proportion of eligible patients who are subsequently enrolled and the number of patients recruited per site per month. The number of patients screened, number/per cent of patients approached, number/per cent of patients excluded after screening/approach and the number/per cent of patients providing consent will be assessed. Reasons for declining participation will be sought where possible, and the appropriateness and practicalities of the chosen eligibility criteria will be explored. The number/per cent of women enrolled prior to initial surgery compared with following neoadjuvant chemotherapy will be assessed. The timing of randomisation in relation to operation start will be recorded to assess the practicalities of randomising as late as possible, in particular, what proportion are randomised on the day of surgery itself.

Use of ICS blood and donor blood will be recorded for both arms, partly to assess intervention fidelity but also to obtain an estimate of the proportion of people in the

control arm that actually require donor blood. Reasons for non-use of ICS blood and/or use of donor blood in the ICS arm will be recorded.

Since the intervention takes place in the operating theatre, it is unlikely that any participant will withdraw from intervention following randomisation. Attrition will be assessed by examining the number of participants lost to follow-up at any subsequent point in the study period. Reasons for discontinuation of follow-up will be sought from participants.

The success of blinding of allocation for participants and outcome assessors will be assessed by asking both the participant and research nurse to guess the allocation (including 'unsure') at the 30-day postoperative follow-up and comparing the responses with the actual allocation.

### Clinical outcomes

In the later, definitive trial, our primary outcome is likely to be either mortality or cancer recurrence, both of which are unlikely to occur in the time available in this feasibility study. Therefore, while readily accessible, these data will not be collected here. Other measures proposed for the later trial will be collected in this feasibility study at baseline and perioperatively, with follow-up at 30 days and 6 weeks postoperatively. Participants recruited at an early stage of the study will also be followed up at 4.5, 7.5 and 10.5 months postoperatively as time allows (figure 1). Clinical outcomes include:

► Inadvertent visceral injury (bladder, bowel, ureters, blood vessels, nerve)
► Return to theatre within 48 hours
► Surgical site infection (see online supplementary appendix 4) within 30 days

► Thromboembolic complications (DVT, PE) within 30 days
► Number and nature of adverse events
► Amount of donor blood given (total and ≤24 hours postsurgery)
► Length of hospital stay
► Resource use
► Generic quality of life (QOL) measure: EQ-5D-5L
► Cancer-specific QOL measure: EORTC QLQ-C30 (Version 3.0) (confirmed cancer only)
► Ovarian cancer QOL measure: EORTC QLQ-OV28 (confirmed cancer only)

### Data management

Each participant will be allocated a unique trial number on consenting to the study and will be identified in all study-related documentation by her trial number and initials. A record of names and addresses linked to participants' trial numbers will be maintained by the research nurse at each site for administrative purposes, and stored securely.

### Data collection

Data collected by the research team (table 1) up to 30 days postoperatively will be recorded on study-specific data collection forms (CRFs), usually by a research nurse. All data not routinely captured during the hospital admission but recorded straight into the CRF will be classified as source data. Participant self-completion questionnaires at baseline will be completed during a face-to-face meeting with a research nurse, following written informed consent. The research nurse will return completed CRFs and baseline questionnaires to the CTU. Subsequent self-completion questionnaires (6 weeks postoperatively

**Table 1** Trial schedule

| | Preoperative | | Operation and perioperative data collection | | | | |
| | | | Postoperative follow-up | | | | |
| | | | 1 | 2 | 3† | 4† | 5† |
| | | | 30 days postop | 6 weeks postop | 3 months after follow-up 2 | 6 months after follow-up 2 | 9 months after follow-up 2 |
| | Screen | Baseline | | | | | |
| Screen/eligibility | x | | | | | | |
| Consent | | x | | | | | |
| Demographics and history | | x | | | | | |
| Randomisation | | x | | | | | |
| EORTC QLQ-C30* | | x | | x | x | x | x |
| EORTC QLQ-OV28* | | x | | x | x | x | x |
| EQ-5D-5L | | x | | x | x | x | x |
| Adverse events | | | x | | | | |
| Resource use questionnaire | | x | | x | x | | |
| Qualitative interviews | | | | x | x | | |

and 3 monthly thereafter as time allows) will be mailed to participants directly from the CTU and returned by participants to the CTU in a prepaid envelope provided. In the event of non-return of a questionnaire, a reminder will be sent from the CTU in the first instance. If there is no response from the two mailings, the CTU will inform the local research nurse who will telephone the participant in order to encourage compliance with follow-up.

## Statistical considerations

Sample size for a feasibility study is necessarily a compromise between the twin assets of precision and efficiency. For any binary 'outcome', our target sample size of 60 will result in a 95% CI of no greater than about +/−12 percentage points, while in a single arm, the target of 30 will have a CI of no more than +/−17 percentage points.

Data analysis will enable the feasibility outcomes to be addressed in order to inform a decision about proceeding to a definitive trial. Data will be presented in accordance with the extension to the Consolidated Standards of Reporting Trials statement for pilot and feasibility studies. They will detail the numbers of patients that were approached, the number that were eligible and the number providing consent. Likewise, compliance rates at all stages will be presented, including the numbers of questionnaires completed at each stage and more generally the completeness of data on all outcomes at each time point. Participating patients' characteristics (demographics, comorbidities, clinical details) will be summarised and, where possible, compared with the overall population of relevant patients to explore possible factors associated with participation. Where possible, the reasons will be ascertained for potentially eligible patients not being approached to consider participation.

Descriptive data on the clinical outcomes will be presented by trial arm, using appropriate measures of central tendency and variation for continuous measures and numbers/percentages for categorical measures. No formal statistical tests will be conducted.

## Qualitative study

A qualitative evaluation will assess the acceptability of the intervention to women taking part in the study, in particular attitudes towards reinfusion of salvaged blood and transfusion of donor blood. The study will also gain an understanding of the women's experience of taking part in the research processes of the TIC TOC study and what influenced their decision to take part. Following surgery, up to 20 women from across all centres will be asked to take part in individual face-to-face or telephone semistructured interviews using a topic guide that has been developed with patient and public involvement (PPI) involvement (see online supplementary appendix 5). Purposive sampling techniques will ensure a range of women are selected according to centre, education, age, ethnicity, socioeconomic status and social support.

As the trial schedule allows, the same women will be approached to take part in a brief telephone interview 3 months after the first interview. The purpose of the second interview is to determine participants' perceptions about the follow-up research processes and ask their opinion about whether anything should change in a full trial. Surgeons from each centre will also be invited to participate in one brief telephone interview each to understand the issues considered in deciding whether to offer women the opportunity to take part in the study.

The qualitative data will be managed using computer software such as Nvivo 11 and thematically analysed.[37 38] The researcher will ensure accuracy of the transcription and read the transcript several times to become immersed in the data, noting initial thoughts and ideas. Codes will be assigned to extracts of the data relevant to the project. Codes with similar meaning will be grouped together in themes. Using constant comparison techniques across the transcripts' themes looking for similarities and differences, the themes will be reviewed and refined. Extracts from the data will be used in the final report. Reflexive research memos will be used as an audit trail of the analysis procedure.[39] A second qualitative researcher will conduct an independent analysis of a subset of six transcripts before the researchers meet to discuss and agree the findings. Findings will also be presented to the study's patient advisory group for discussion. Any significant differences of opinion will be discussed with the chief investigator. A model may be developed to explain the factors affecting recruitment and retention to the trial to inform development of the research processes required in any future full trial.

## Economic data and analyses

A definitive study will include a within-trial economic evaluation to compare costs and health outcomes of ICS versus donor blood within the time frame of the study and a decision analytic model to extrapolate any future health benefits and costs to the lifetime of the participant. The evaluations will primarily be in relation to quality-adjusted life-years and will take a health and social perspective on costs, in accordance with National Institute of Health and Care Excellence (NICE) guidelines.[37] Secondary analyses will take place in relation to important clinical outcomes of interest for the definitive trial such as deaths averted and disease-free progression. This study aims to test the feasibility of collecting enough resource use and outcome data to perform the future economic evaluations.

Data collection tools will be prepared and refined with a view to undertaking the two planned economic evaluations within the future study. These evaluations will take on a health and social care payer perspective. Should participant-reported resource use data allow, the future within-trial economic evaluation will take on a societal perspective on costs in secondary analyses, to further capture the burden to participants, carers and society. The parameters for the lifetime economic decision model (costs, outcomes and probabilities of outcomes to occur) will be informed by the within-trial economic results. If feasible, costs from a societal perspective may be included in the lifetime economic decision model as well.

Resources will be collected from several sources. In the immediate postoperative period, research nurses will record resources pertaining to the participant's surgery and subsequent hospital stay. Where possible, research staff will also review participants' medical notes at 4.5 months postoperatively to collect hospital contacts following initial discharge (ie, rehospitalisations, outpatient and emergency visits). Participant-completed resource use questionnaires will be administered at both 6 weeks and 4.5 months postoperatively (where the trial schedule allows) to collect other resources used. These questionnaires will be delivered by post and include questions related to inpatient and outpatient hospital visits; community-based services such as general practice doctor and nurse contacts, physiotherapy, occupational therapy and other community contacts; use of personal social services such as home care workers and social workers; privately paid therapies and expenses; time off work and lost leisure; and informal care required from family and friends. Completion rates, missing data and the method of administering questionnaires will be reviewed to identify potential problems with data collection methods and to seek solutions to minimise participant/staff burden if required. We will report frequency, mean and SD of resources used by trial arm to explore potential cost drivers for the main study.

The EuroQoL EQ-5D questionnaire will capture generic QOL differences between the trial arms. In a recent study of EQ-5D valuation sets, the 3 L and 5 L versions of the EQ-5D produced substantially different estimates for cost-effectiveness[40] and prompted NICE to issue a position statement in August 2017 to recommend the future use of the 3 L version.[41] In this study, we will use the mapped utility scores from the 3 L to the 5 L version using the Van Hout algorithm[42] for the UK population, as recommended by the NICE statement. We expect to use the 3 L version in the future study and not proceed with the study of the distribution properties produced by the 5 L version scores in this feasibility study.

## PPI and engagement

The study has benefitted from its inception from an enthusiastic patient advisory group. The aim of PPI in the study is to ensure that the trial is equitable and acceptable to the women taking part by embedding the women's experiential expertise of cancer throughout the trial design and processes. The group comprises six women aged between 50 and 80 years, who have experienced a cancer diagnosis and are living in Cornwall. However, one member is formerly from Gateshead, where she was treated for her cancer, so she is able to bring her experience of the patient pathway to inform the trial processes across the sites. Another member and coapplicant is the founder of PANTS cancer charity in Cornwall.

The PPI work is undertaken using a predominately collaborative approach with engagement functions embedded within it. The members worked with the research team on the research design and in particular

the patient approach, providing input into the grant application, language, content and layout of the participant documentation. The group have worked on the qualitative interview topic guide content and are also working with the qualitative researchers on analysis of the participant interview transcripts. The members are fully integrated into the team and regularly attend the trial management meetings, as well as provide advice and suggest solutions to problems encountered during the trial.

The members will attend patient and public events and conferences to engage with other members of the public and professionals and share their experience of supporting and being part of the design and management of research. They will also work together with the wider research team to prepare a lay summary of the findings and on other communications such as website, Twitter and Facebook articles.

All members of the research team contribute to the training and support of the PPI members. The mechanisms to achieve these are multifactorial and include specific discussion around methodology and trial processes in PPI meetings, explaining the terminology in lay language, providing information, such as the INVOLVE jargon buster sheet, and conducting workshops for specific tasks (eg, poster development), as well as signposting to other resources such as the INVOLVE website.

## ETHICS AND DISSEMINATION

The results of this feasibility study will be published in peer-reviewed journals and presented at relevant national/international conferences and to patient groups. Participants of the trial will be sent a summary of the findings and these will also be disseminated via the pantscancer.org charity, Target Ovarian Cancer charity and participating NHS Trusts' websites.

## DISCUSSION

Research has shown that donor blood transfusions have been associated with poorer outcomes including increased mortality, wound, pulmonary and renal complications; this has been ascribed to TRIM[9] which is a transient depression of the immune system following transfusion with blood products. The Cochrane meta-analysis of randomised trials estimated perioperative ABT to be associated with increased risk of recurrence with OR of 1.42 (95% CI 1.20 to 1.67) in surgery for colorectal cancer.[43] Long-term results from a clinical trial suggest that this effect of ABT is persistent.[44 45] This led to the suggestion of introducing measures that would help limit the use of ABT.[12]

Patient blood management is an evidence-based patient-tailored approach aimed at reducing the need for ABT by managing anaemia, perioperative blood conservation, surgical haemostasis and drug use.[46] Perioperative

blood conservation measures include interventions such as the administration of agents to diminish blood loss (eg, tranexamic acid, fibrin sealant), agents that promote red blood cell production (eg, erythropoietin) and techniques for reinfusing a patient's own blood including cell salvage.[28] Previous randomised and non-randomised studies have provided evidence that the use of ICS can reduce the need for ABT.[9] A systematic review of 75 randomised trials highlighted that salvaged blood reinfusion reduced the rate of exposure to ABT by 38% (relative risk, 0.62; 95% CI 0.55 to 0.70).[47] However, concern exists that blood collected by ICS might result in reinfusion of tumour cells and subsequent distant metastases thus limiting the use of cell salvage across oncological specialties. However, in patients undergoing surgery for a gynaecological malignancy, the use of a leucocyte depletion filter was shown to be effective in eliminating viable nucleated malignant cells from the returned blood during collection, processing and leucofiltration.[27] Similarly, in vitro work shows that depletion filters are highly efficient at removing malignant cells, leading to removal rates of between 80% and 100%.[25 26]

Patients with primary or metastatic cancer are known to have CTCs in the blood. The concentration of CTCs varies widely depending on tumour type and stage of disease.[31] There is evidence from a range of different cancer surgeries that operative manipulation of tumour during surgery leads to peripheral blood concentrations of malignant cells many times higher than could be attained with cell salvage alone.[31 32 48]

There is emerging evidence suggesting that far from compromising outcomes, intraoperative autologous transfusion is associated with improved outcomes in surgery for other gynaecological cancers such as cervical cancer. Several studies in patients with early stage (I–IIA) cervical cancer report that intraoperative autologous transfusion significantly reduces the need for donor blood transfusion, without compromising survival or postoperative complication rates.[30] In addition, no distant recurrences have been reported.[30] However, most of the evidence on the use of salvaged blood in cancer surgery is based on retrospective and observational studies. These studies are insufficient to draw any definitive conclusions regarding adverse events related to a particular intervention in the presence of multiple confounding factors. Therefore, in order to mitigate for confounding factors, a large well-designed RCT is required.[49] Our trial provides new evidence in the use of cell salvage in ovarian cancer surgery and will add to a more general evidence base informing the use of ICS in other areas, in particular other cancers.

**Author affiliations**

[1]Gynaeoncology, Royal Cornwall Hospitals NHS Trust, Truro, UK

[2]Medical School, University of Exeter, Exeter, UK

[3]Research, Development and Innovation, Royal Cornwall Hospitals NHS Trust, Truro, UK

[4]NIHR research and Design Services (South West), NIHR South West Research Design Service, Plymouth, UK

[5]Gynaecology, Royal Cornwall Hospital NHS Trust, Truro, UK

[6]Bristol Medical School, University of Bristol, Bristol, UK

[7]Royal Cornwall Hospitals NHS Trust, Truro, UK

[8]Peninsula Clinical Trials Unit, Plymouth University, Plymouth, UK

[9]Department of Anaesthesia, Royal Cornwall Hospitals NHS Trust, Truro, UK

**Acknowledgements** The authors are grateful for the support of the study sponsor (Royal Cornwall Hospitals NHS Trust) and the South West Peninsula NIHR Clinical Research Network. We are also indebted to the members of our PPI group for their continued support of the trial. In addition, the authors would like to thank the following members of the TIC TOC trial team: Mr S Chatopadhyay, Mr G Hughes, Mr R Naik and Dr P Ricketts.

**Contributors** All authors except NF and JP were coapplicants on the NIHR RfPB grant application and as such were involved in the design of this feasibility study. All authors contributed to successive drafts of this paper. KG is the chief investigator, provided clinical expertise and was responsible for conception and design of the study as well as drafting and revising of the article. NF was responsible for the first draft of this paper. CP contributed to study design. AB contributed to study design and trial management. PE is the trial statistician and provided expertise in the overall design of the trial. JF provided expertise in cell salvage and drafted the cell salvage protocol. AL provided clinical expertise and helped with the design of the study. EMRM was responsible for the design and analysis of the economic evaluation component. JP contributed to the study design and coordinated the PPI input. CR provided anaesthetics and cell salvage expertise. PJV is the trial manager, responsible for overseeing the day-to-day running of the trial. JW was responsible for the design and conduct of the qualitative study.

**Funding** This work was supported by the National Institute for Health Research (NIHR) Research for Patient Benefit (RfPB) programme, grant number PB-PG-1014-35005.

**Disclaimer** The views expressed are those of the author(s) and not necessarily those of the NHS, the NIHR or the Department of Health.

**Competing interests** None declared.

**Patient consent** Not required.

**Ethics approval** South West–Exeter Research Ethics Committee.

**Provenance and peer review** Not commissioned; externally peer reviewed.

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
