## [Reviewer comments · BMJ Open]

ARTICLE DETAILS

TITLE (PROVISIONAL)	Trial of intraoperative cell salvage versus transfusion in ovarian cancer (TIC TOC): protocol for a randomised controlled feasibility study
AUTHORS	Galaal, Khadra; Lopes, Alberto; Pritchard, Colin; Barton, Andy; Wingham, Jennifer; Marques, Elsa; Faulds, John; Palmer, Joanne; Vickery, Patricia; Ralph, Catherine; Ferreira, Nicole; Ewings, Paul

VERSION 1 – REVIEW

REVIEWER	Steven Frank Johns Hopkins Medicine, Baltimore, Maryland, USA
REVIEW RETURNED	20-May-2018

GENERAL COMMENTS	This randomized trial protocol is submitted for review for possible publication before the trial begins. Therefore, there are no results or conclusions, which is to be expected. The proposed study is to compare allogeneic blood transfusion to intraoperative cell salvage for ovarian cancer surgery and this proposal is for a feasibility pilot study. 60 adult females will be randomly assigned to one of these two treatments, and effectiveness and cost-effectiveness of cell salvage vs. allogeneic blood will be compared. If the feasibility study succeeds, the authors plan a large RCT to assess important outcomes like economic ones and cancer recurrence and mortality. Overall, I understand the need for this pilot study to determine feasibility, because this may be a very difficult study to conduct. For example, there is a strong opinion among healthcare professionals, that cancer surgery is a contraindication to cell salvage. The authors cite 4 or 5 studies suggesting that salvage is safe, however many clinicians remain skeptical, nonetheless. This is actually a good reason to do feasibility, because patients who read on the internet will also be very skeptical, and there will be many of these patients who will be reading on the internet, guaranteed. Another concern I have is patients getting different variations of treatment. For example, in the salvage group they could receive: 1) no blood at all, 2) allogeneic blood only, 3) salvaged blood only, or 4) both salvaged and allogeneic blood. This is because not all salvaged cases yield enough shed blood to be processed and returned to the patient. And not all salvaged blood is enough to satisfy their needs, and the patients often need more than salvaged blood alone. In the allogeneic group, they could either receive: 1) no blood at all, or 2) allogeneic blood. I am afraid that with 60 patients, with 30 in each group, that these scenarios will yield a lot of variation in treatments, with very small "n"s in the
---

	individual groups. I do not see the authors plan to analyze the data according to assigned treatment or received treatment, or both ways. Page 7 – LM 56 – On the telephone follow up for adverse events, what are these adverse events? I don't see what events are being assessed. Page 9 – LM 4 – I think the randomization should be stratified on primary surgery vs. post chemotherapy surgery. This way there will be less chance of confounding by this factor. Page 10 – The investigators should incorporate some form of Hb trigger into the protocol for transfusion, for allogeneic blood. Even if it is just a guideline and not absolute. Otherwise the use of banked blood could be different in the two groups due to treatment bias. Page 10-11 – What about partially filled cell salvage bowls? Will they be processed and returned to the patients? This is controversial due to incomplete washing and potential for lower quality salvaged product. What size bowls will be chosen and by what criteria? A smaller bowl will ensure more complete washing and more yield back to the patients. Page 11 – LM 48 – What is “resource use”? Does this mean total costs? If so, cost of what? Page 13 – Will the same research nurse keeping the record log be tracking postop outcomes, because this would break the blindedness. Page 16 – LM 22 – In the bigger study after the pilot, how will disease free progression be assessed? Imaging studies or biochemical tests? Are there any biochemical markers for recurrence of ovarian Ca? Page 31 – The questions posed in appendix 3 have the potential to generate answers that are not quantitative and not analyzable. For example, “what did you understand about reintroducing your own blood?” or “what did you understand about the study?”
--	--

REVIEWER	A/ Prof Naresh Kumar National University Health System, Singapore, Orthopaedci Surgery
REVIEW RETURNED	13-Jun-2018

GENERAL COMMENTS	As mentioned in the attached document. In addition: 5. They can be a little more elaborate about their research ethics approval. 7. Unfortunately, I am not an expert in statistics. It is appropriate as per my knowledge. I would however rely on the statistician employed by the journal to look at the statistics more critically. 8. We have suggested some references to be changed, in the accompanying pdf document. - The reviewer provided a marked copy with additional comments. Please contact the publisher for full details.
--

VERSION 1 – AUTHOR RESPONSE

Reviewer 1

Page 7– LM 56 – On the telephone follow up for adverse events, what are these adverse events? I don't see what events are being assessed.

Response: These are adverse events related to the procedure (surgery) or to the blood replacement therapy (allogeneic transfusion).

Page 9 – LM 4 – I think the randomization should be stratified on primary surgery vs. post chemotherapy surgery. This way there will be less chance of confounding by this factor.

Response: Thank you, ideally that is going to be addressed in the phase 3 randomised trial

Page 10 – The investigators should incorporate some form of Hb trigger into the protocol for transfusion, for allogeneic blood. Even if it is just a guideline and not absolute. Otherwise the use of banked blood could be different in the two groups due to treatment bias.

Response: This is included in the protocol for the study

Page 10-11 – What about partially filled cell salvage bowls? Will they be processed and returned to the patients? This is controversial due to incomplete washing and potential for lower quality salvaged product. What size bowls will be chosen and by what criteria? A smaller bowl will ensure more complete washing and more yield back to the patients.

Response: we will use the blood in partially filled bowls as per protocol

Page 11 – LM 48 – What is “resource use”? Does this mean total costs? If so, cost of what?

Response: This is explained in the 'Economic data and analyses' section, with specific detail on Page 17, lines 1-10.

Page 13 – Will the same research nurse keeping the record log be tracking postop outcomes, because this would break the blindedness.

Response: No

Page 16 – LM 22 – In the bigger study after the pilot, how will disease free progression be assessed? Imaging studies or biochemical tests? Are there any biochemical markers for recurrence of ovarian Ca?

Response: Disease recurrence is assessed by imaging with CT scan and tumour marker test CA125, this is evaluated in addition to any symptoms the patients report

Page 31 – The questions posed in appendix 3 have the potential to generate answers that are not quantitative and not analyzable. For example, “what did you understand about reintroducing your own blood?” or “what did you understand about the study?”

Response: Appendix 3 is the topic guide for participant interviews which will be analysed using qualitative methods.

Reviewer 2

Page 3, line 13 (first mention): reviewer suggested changing 'donor' blood to 'allogeneic blood'.

Response: We used the description of donor blood transfusion following advice from our PPI group.

Therefore, we will stick with our description, however, we have added the term allogeneic in brackets.

Page 3, line 37: 'We are always worried about tumour recurrence and disease progression, which may be related to immune suppression due to transfusion. I felt that there should be a method of studying them'.

Response: Agree with your comment, however, this is a feasibility study and this aspect will be investigated in the full randomised trial.

Page 4, line 10: Should this not be able to cover long term outcomes like disease progression?

Response: Agree with your comment, however, this is a feasibility study and this aspect will be investigated in the full randomised trial.

Page 5, line 12: Sentences are unclear. Kindly define what surgical success and radiological success are; and how each of them are determined.

Response: changes made in the text to clarify.

Page 6, line 14: 'Why choose the reference of oesophageal cancer, where the survival itself is very low? This reference could be a better one. Kindly revise'.

Response: reference changed.

Page 6, line 34: 'We suggest two recent references as follows: Flow Cytometric evaluation of the use of Salvaged Blood in Metastatic Spine Tumour Surgery – Naresh Kumar, R. Lam, A.S. Zaw, R. Malhotra, Tan J.H. Jonathan, G. Tan, T. Setiobudi. Ann Surg Oncol, 2014 Dec; 21(13): 4330-5. [DOI: 10.1245/s10434-014-3950-9]. Epub 2014 Jul 29. PMID- 25069862.

Intraoperative cell salvage in metastatic spine tumour surgery reduces potential for reinfusion of viable cancer cells. Naresh Kumar, A.S. Zaw, B.L. Khoo, S. Saminathan, Z. Jiang, G. Singh, C.T. Lim, B.L. Thiery. EurSpine J. 2016. Dec;25(12):4008-4015. [DOI: 10.1007/s00586-016-4478-4]. PMID: 26951173'

Response: references now included.

Page 7, line 16: 'There is still an unfounded fear regarding re-infusion of salvaged blood in oncological patients. Hence blinding may still be controversial'.

Response: agree, blinding was only for the patient and outcome assessor.

Page 7, line 54: 'Note the comments in the 'Aims and Objectives' section

Response: Addressed above

Page 8, line 1: 'Kindly note the comments provided in the abstract'

Response: Addressed above.

Page 10, line 24: 'Are there any publications to say that the use of leucodepletion filter (LDF) reduces the amount of ICS blood dramatically? Is LDF necessary or could this study have been done without using LDF? There is some evidence in literature to say that ICS blood can be safely re-infused to patients without the use of LDF'.

Response: We felt with this being the first trial on using ICS in ovarian cancer, LDF filter usage would help us in improving its acceptability, especially among surgeons

There is evidence that LDF removes tumour cells in gynaecology, please check this reference:

Catling S WS, Freitas O, Rees M, Davies C, Hopkins L. Use of a leucocyte filter to remove tumour cells from intra-operative cell salvage blood. Anaesthesia 2008;63(12):1332-8.

Page 13, line 20: 'Kindly mention the appropriate abbreviation - CRFs stand for 'Case Report Forms'?

Response: This was expanded when first mentioned in the text on page 10, line 28.

Page 18, line 44: Kindly elaborate a few sentences on whether an Institutional Review Board approval was sought for.

Response: This is already stated on page 2, line 46.

Page 19, line 1: 'Kindly mention references to support this'.

Response: done and reference included.

Page 19, line 37: 'Kindly check the reference. This sentence is not quoted properly. Should it be reference 11?'

Response: done and reference added.

VERSION 2 – REVIEW

REVIEWER	Steven M. Frank MD Johns Hopkins University, USA
REVIEW RETURNED	08-Aug-2018
GENERAL COMMENTS	The type of leukoreduction filter should be specified as there are different kinds.

VERSION 2 – AUTHOR RESPONSE

In this feasibility study it is important that a leucodepletion filter is used in the process of returning salvaged blood but it does not matter which make of filter is used. The manuscript text has been amended to clarify this.